# Nanoparticle-Mediated Delivery of STAT3 Inhibitors in the Treatment of Lung Cancer

**DOI:** 10.3390/pharmaceutics14122787

**Published:** 2022-12-13

**Authors:** Qiyi Feng, Kai Xiao

**Affiliations:** 1Precision Medicine Research Center, Sichuan Provincial Key Laboratory of Precision Medicine, National Clinical Research Center for Geriatrics, West China Hospital, Sichuan University, Chengdu 610041, China; 2Frontiers Science Center for Disease-Related Molecular Network, West China Hospital, Sichuan University, Chengdu 610041, China

**Keywords:** lung cancer, STAT3, nanoparticles, drug delivery, cancer stem cells

## Abstract

Lung cancer is a common malignancy worldwide, with high morbidity and mortality. Signal transducer and activator of transcription 3 (STAT3) is an important transcription factor that not only regulates different hallmarks of cancer, such as tumorigenesis, cell proliferation, and metastasis but also regulates the occurrence and maintenance of cancer stem cells (CSCs). Abnormal STAT3 activity has been found in a variety of cancers, including lung cancer, and its phosphorylation level is associated with a poor prognosis of lung cancer. Therefore, the STAT3 pathway may represent a promising therapeutic target for the treatment of lung cancer. To date, various types of STAT3 inhibitors, including natural compounds, small molecules, and gene-based therapies, have been developed through direct and indirect strategies, although most of them are still in the preclinical or early clinical stages. One of the main obstacles to the development of STAT3 inhibitors is the lack of an effective targeted delivery system to improve their bioavailability and tumor targetability, failing to fully demonstrate their anti-tumor effects. In this review, we will summarize the recent advances in STAT3 targeting strategies, as well as the applications of nanoparticle-mediated targeted delivery of STAT3 inhibitors in the treatment of lung cancer.

## 1. Introduction

Lung cancer is one of the most common malignancies and the leading cause of cancer-related deaths, with an estimated 2.09 million newly diagnosed cases in both genders and 1.76 million deaths worldwide [1]. Small cell lung cancer (SCLC) and non-small cell lung cancer (NSCLC) are the two main pathological types of lung cancer, and NSCLC accounts for approximately 80% to 85% of all cases. NSCLC has three main subtypes, including adenocarcinoma, squamous cell carcinoma, and large cell carcinoma [2]. Treatment of lung cancer varies greatly depending on the type and stage of the disease, including traditional chemotherapy, radiotherapy, targeted therapy, and immunotherapy. In recent years, targeted therapy has emerged as an important therapeutic strategy for the management of NSCLC. For example, epidermal growth factor receptor (EGFR)-tyrosine kinase inhibitors (TKIs) and anaplastic lymphoma kinase (ALK)/c-ros oncogene 1 (ROS1) inhibitors have replaced chemotherapy as the first-line treatment of lung cancer [3]. Although targeted therapies are initially effective, acquired drug resistance is usually inevitable due to cancer-driven genetic alterations, epigenetic alterations, and tumor heterogeneity [4]. Immunotherapy, such as immune checkpoint blockade (ICB), is a new treatment strategy that may improve the survival of lung cancer. However, the clinical benefits of ICB in the treatment of advanced lung cancer have proven to be limited and unsatisfactory, with an overall response rate (ORR) of approximately 10–20% [5]. Therefore, there is an urgent need to identify alternative strategies to improve existing treatments or to provide new treatments.

Signal transducer and activator of transcription 3 (STAT3), a key component of the Janus kinase (JAK)–STAT pathway, is found aberrantly activated in the majority of NSCLC patients [6]. STAT3 can be phosphorylated by multiple cytokines, interferons, and growth factors, and the activated STAT3 can be transferred to the nucleus and bind to the promoter and enhancer regions of target genes to modulate gene transcription [7]. Multiple studies have proven that increased phosphorylated STAT3 (pSTAT3) is usually associated with cell proliferation, invasion, and angiogenesis, leading to tumor progression, metastasis, drug resistance, and immune escape [8]. Moreover, the existence of a subpopulation of cells in tumors called cancer stem cells (CSCs) or tumor initiating cells (TICs), responsible for drug resistance, tumor metastasis, and relapse, is also reported to correlate with increasing levels of STAT3 activation [9]. Thus, targeting the STAT3 signaling pathway has emerged as a promising therapeutic strategy for lung cancers.

A variety of nanocarriers have been recently developed to deliver poorly water-soluble drugs, thus improving their bioavailability and targeting capability. Nanoparticles (NPs) with particle sizes of 10–100 nm can increase drug enrichment in tumor tissue and reduce its distribution in normal tissue through enhanced permeability and retention (EPR) effect and active targeting strategies (e.g., decoration of specific ligands) [10]. NPs-based drug delivery systems, such as inorganic NPs, polymeric NPs, micelles, dendrimers, and liposomes, have shown great potential in the diagnosis, imaging, and treatment of cancer. Some of these nanocarriers have been successfully used in the clinic for drug delivery, such as Abraxane^®^ (albumin-bound paclitaxel) and Doxil^®^ (liposomal doxorubicin).

In this review, we discuss and summarize the importance of the STAT3 signaling pathway in the progress of lung cancer and the recent progress in STAT3 targeting strategies. In addition, we also introduce the potential application of NPs-based targeted delivery of STAT3 inhibitors in the treatment of cancers, including lung cancer.

## 2. Role of STAT3 in Lung Cancer Cells

It is believed that multiple signaling pathways are related to the initiation and metastasis of lung cancer. STAT3, one of the numerous transcription factors among those signal pathways, is closely related to the occurrence and development of cancer and is considered a promising target for cancer treatment (Figure 1). Among the seven members of the STAT family (STAT1, STAT2, STAT3, STAT4, STAT5a, STAT5b, and STAT6), STAT3 has attracted much attention because of its vital function in cell differentiation [11]. The structure of STAT3 is comprised of six functionally conserved domains, including the amino-terminal domain (NH2), coiled-coil domain (CCD), DNA-binding domain (DBD), linker domain, SRC homology 2 (SH2) domain, and carboxyl-terminal transactivation domain (TAD) [12]. Among them, the SH2 domain plays an important role in STAT3 activation by engaging the dimerization of two phosphorylated STAT3 monomers [13]. The activation of STAT3 can be induced by nearly 40 different polypeptide ligands, such as cytokines, growth factors, carcinogens, and environmental stress [11]. It is mainly activated by the direct phosphorylation of tyrosine (705) and serine (727) residues to form a STAT3 dimer, which is subsequently translocated into the nucleus to regulate target genes [12]. Under normal circumstances, the activation of STAT3 is negatively regulated by suppressors of cytokine signaling (SOCS) proteins, protein inhibitors of activated STATs (PIAS) proteins, and protein tyrosine phosphatases (PTPases) and is involved in mammary gland development and embryogenesis through interferon-gamma (IFN-γ) signaling, the cell cycle, and apoptosis [13]. However, the constitutive activation of STAT3 is Involved in many cellular processes, including survival, proliferation, invasion, angiogenesis, metastasis, and immunosuppression, all of which favor tumor initiation and progression [12,14].

There is evidence that abnormally activated STAT3 has been found in a variety of cancers, including lung cancer, which leads to the inactivation of apoptotic pathways and the resistance of cancer cells to radiotherapy and chemotherapy [15,16,17]. The aberrant activation of STAT3 is usually associated with poor tumor differentiation, advanced clinical stage, lymph node metastasis, and drug resistance of lung cancer [13]. Recent studies have shown that microRNAs could promote proliferation, migration, invasion, anti-apoptosis, and angiogenesis of lung cancer cells by activating STAT3 [18,19,20]. Meanwhile, microRNAs could also inhibit epithelial–mesenchymal transition (EMT) and reverse drug resistance by inhibiting the STAT3 signaling pathway, thereby suppressing tumor growth and improving the therapeutic effect of chemotherapeutic drugs [21,22]. In addition, alterations of STAT3 activity using genetic and/or pharmacological methods may be effective in regulating immunosuppression in lung cancer cells, affecting the outcome of cancer immunotherapy [23]. Studies have shown that dysregulated IL-6 or JAK2 can reprogram the STAT3 pathway in metastatic tumor cells, induce recruitment of myeloid-derived suppressor cells (MDSCs) and polarized macrophages, and repress the infiltration of CD8+ T cells to evade host immunity in NSCLC [24,25,26]. On the contrary, inhibition of STAT3 can downregulate the expression of programmed cell death 1 ligand 1 (PD-L1) and enhance the infiltration of T cells into tumor tissues, exhibiting synergistic effect in combination with programmed cell death protein 1 (PD-1)/PD-L1 blockade [27,28].

It has been reported that STAT3 activation could not only stimulate a series of cascades associated with the development and progression of lung cancer but also serve as a key mediator to regulate the characteristics of lung cancer stem cells (LCSCs) [29]. Studies have shown that STAT3 can be phosphorylated by receptor tyrosine kinases (RTKs) such as EGFR and insulin-like growth factor 1 receptor (IGF1R), or non-RTKs such as c-Src and JAK, thereby maintaining the EMT-associated, CSC-like properties in cells [30,31,32]. For example, it was found that c-Src/IGF1R-mediated STAT3 activation was regulated by Tescalcin (TESC), which enhanced the expression of acetaldehyde dehydrogenase 1 (ALDH1) and thus reinforced CSC-like and radiation-resistant properties [11]. Another study showed that the elevation and activation of the Aryl hydrocarbon receptor (AhR) in NSCLC cells could induce JAK2/STAT3 phosphorylation, while the inhibition of JAK2/STAT3 signaling by pharmacologic approaches can attenuate the AhR-mediated stemness effects of NSCLC cells, indicating that the JAK2/STAT3 pathway plays a vital role in AhR-regulated NSCLC stemness [14], and targeting STAT3 has become a promising strategy for lung cancer treatment.

It is worth noting that STAT3 has also been reported to have a potential role in tumor suppression in addition to promoting the development of lung cancer. The activation of STAT3 in mice may reduce the occurrence and malignant progression of Kras (G12D)-driven lung adenocarcinoma by controlling the expression of interleukin 8 (IL-8) induced by nuclear factor kappa-B (NF-κB), and inhibiting IL-8-mediated myeloid tumor infiltration and tumor vascularization [33]. Another study indicated that STAT3 might prevent the initiation of lung cancer by maintaining pulmonary homeostasis under oncogenic stress, but it may also facilitate the progression of lung cancer by promoting the growth of cancer cells at the same time [34]. Such contradictory conclusions make targeting STAT3 a complicated process, and the role of STAT3 in tumor initiation and progression must be carefully studied.

## 3. Therapeutic Strategies Targeting STAT3

Given the important role of abnormally activated STAT3 in the occurrence and progression of lung cancer, targeted inhibition of STAT3 may become an effective therapeutic strategy for lung cancer. The ways targeting STAT3 may be direct or indirect, mainly including non-oncology drugs, natural products and derivatives, small-molecule inhibitors, nucleotide-based therapeutics, and agents regulating STAT3 upstream genes (Table 1).

### 3.1. Non-Oncology Drugs

Several known drugs that were not initially used in cancer treatment have shown significant anti-cancer activity by targeting the STAT3 signaling pathway. Antiparasitic drugs have been demonstrated to successfully inhibit the activation of the STAT3 pathway, selectively impairing the growth of lung cancer cells and eliciting lethal effects both in vitro and in vivo [28,35,36,37]. Moreover, recent studies have shown that niclosamide (an antihelminthic drug) and its derivative (HJC0152) enhance the efficacy of immunotherapy in NSCLC by blocking the binding of pSTAT3 to the PD-L1 promoter, thus downregulating the expression of PD-L1 [28,35]. Other classical drugs exhibited anti-cancer effects by direct or indirect targeting STAT3. A recent study reported that high doses of acetaminophen (AAP), a commonly used antipyretic and analgesic drug, exhibited anti-CSCs activity in lung cancer and melanoma cells by directly binding STAT3 with an affinity in the low micromolar range [38]. In addition, dihydroartemisinin (DHA), a semisynthetic derivative of the herbal antimalarial drug artemisinin, can effectively inhibit STAT3 phosphorylation, downregulate myeloid cell leukemia-1 (Mcl-1) and survivin, and enhance ABT-263 (Bcl-2 inhibitor)-induced cytotoxicity [39]. Therefore, non-oncology drugs capable of targeting STAT3 may have therapeutic potential in lung cancer.

### 3.2. Natural Compounds and Derivatives

Some researchers have explored the effects of natural compounds to develop anti-CSCs therapy. Curcumin (CUR) and its derivatives, one of the most commonly used natural compounds for cancer therapy, have been reported to inhibit the phosphorylation of STAT3 and its downstream genes, thereby inhibiting angiogenesis and tumor growth [40,41]. Terpenoids, widely distributed in nature, have been extensively studied for their biological activities, including anti-cancer effects. For example, Cucurbitacin I (a typical terpenoid) might inhibit the phosphorylation of STAT3 and enhance the phosphorylation of STAT1 in A549 lung adenocarcinoma cells by disrupting actin filaments [42]. Another natural triterpenoid extracted from Anemone Raddeana Regel, Raddeanin A, was able to inhibit the expression of pSTAT3 and STAT3, reduce the mitochondrial membrane potential, and promote apoptosis in A549 and H1299 lung cancer cells [43]. Other terpenoids, such as Ginkgolide C (GGC) and eupatolide isolated from Ginkgo biloba (Ginkgoaceae) leaf and *Inula helenium*, respectively, could effectively attenuate the phosphorylation of STAT3 and its upstream kinases, demonstrating significant inhibition of NSCLC tumor growth [44,45].

Although many natural compounds have shown therapeutic effects on lung cancer by inhibiting STAT3 activation, their molecular mechanisms remain to be further studied. Ma et al. investigated the anti-cancer effect and underlying mechanisms of Proscillaridin A (PSD-A), a cardiac glycoside component of *Urginea maritima.* They found that PSD-A was able to inhibit both constitutive and inducible STAT3 activations and reduce STAT3–DNA binding activity, which was associated with increased expression of SH2 domain-containing protein tyrosine phosphatase 1 (SHP-1), decreased phosphorylation of Src, and binding of PSD-A with the STAT3 SH2 domain. This was the first in-depth illustration of the molecular mechanism by which PSD-A inhibits STAT3 [46].

Since the aberrant activation of STAT3 was found to be related to EGFR-TKI resistance in lung cancer, some natural compounds were proved to successfully reduce cell growth and induce apoptosis in EGFR-TKI-resistant NSCLC cells through the suppression of STAT3 activity, showing their potential as novel therapeutics for lung cancer patients with EGFR-TKI resistance [47]. Interestingly, pterostilbene alone, which is isolated from Pterocarpus marsupium (PM) heartwood, was not able to induce anti-proliferative effects in EGFR-mutated NSCLC cells. However, pterostilbene plus osimertinib reversed osimertinib-induced phosphorylation of STAT3, Yes-associated protein 1 (YAP1), and CUB domain-containing protein-1 (CDCP1), abrogating the resistance pathways activated by single osimertinib treatment in EGFR-mutated NSCLC [48].

### 3.3. Small Molecule Inhibitors

Recently, a lot of efforts have been made to develop specific and potent STAT3 small molecule inhibitors. The SH2 domain plays a pivotal role in the STAT3 signaling cascade, so targeting the SH2 domain of STAT3 will prevent the dimerization and transcriptional activity of STAT3 [49]. Therefore, Rangappa et al. performed high-throughput virtual screening using a cheminformatics platform to search for STAT3 inhibitors and identified 2-Amino-6-[2-(Cyclopropylmethoxy)-6-Hydroxyphenyl]-4-Piperidin-4-yl Nicotinonitrile (ACHP) as an effective inhibitor. It was found that ACHP interacted with the SH2 domain of STAT3 and significantly inhibited the phosphorylation of STAT3 at Tyr705, resulting in the apoptosis of NSCLC cells [50]. Another novel STAT3 inhibitor, W2014, characterized by the core structure of imidazopyridine, was able to occupy the sub-pockets of the SH2 domain and bind to STAT3 protein with high affinity. W2014 not only exhibited potent anti-tumor activities but also sensitized drug-resistant NSCLC cells to gefitinib by inhibiting aberrant STAT3 signaling in vitro and in vivo [49]. Moreover, Napabucasin (BBI608) and LL1, two newly developed small molecule inhibitors, were able to significantly inhibit the self-renewal of LCSCs by targeting the SH2 domain of STAT3 [51,52,53,54].

In addition to chemotherapeutic drugs and EGFR-TKI inhibitors, the combination of STAT3 inhibitors and other small molecular inhibitors has also achieved promising therapeutic effects on lung cancer. The STAT3 inhibitors combined with rapidly accelerated fibrosarcoma (RAF) inhibitor or ALK inhibitor effectively suppress the survival of Kras-mutated or ALK-rearranged lung cancer cells by abrogating the activation of mitogen-activated extracellular signal-regulated kinase (MEK)/extracellular regulated protein kinases (ERK) signaling pathway [55,56].

In addition to the SH2 domain, there has been a growing interest in developing specific inhibitors that can target the DBD of STAT3. A new small molecule, (E)-2-methoxy-4-(3-(4-methoxyphenyl)prop-1-en-1-yl) phenol (MMPP), was found to regulate cell cycle and apoptosis-related genes by directly binding to the hydroxyl residue of threonine 456 in the DBD of STAT3, leading to G1-phase cell cycle arrest and apoptosis. In addition, MMPP showed comparable or better anti-tumor activity than docetaxel or cisplatin [57].

Although STAT inhibitors, alone or in combination with other drugs, have achieved promising effects, no STAT3 inhibitor has been approved for the treatment of lung cancer, likely due to their low bioavailability and off-target toxicities. Therefore, the exploration of more efficacious STAT3 inhibitors and other effective STAT3 targeting strategies should be inspired.

### 3.4. Therapeutic RNA Molecules

In recent decades, nucleic acid-based therapeutics have been extensively studied because of their ability to target a broader group of proteins and their ability to be engineered to expand their applicability compared with small molecular drugs [58]. Therapeutic RNA molecules, including antisense oligonucleotides (ASOs), small interfering RNAs (siRNAs), short hairpin RNAs (shRNAs), microRNAs (miRNAs), and long intergenic non-coding RNA (lncRNAs), have been developed for STAT3-targeted lung cancer therapy.

ASOs are short single- or double-stranded oligonucleotides that complement the target mRNA they hybridize to modulate protein expression [58]. Recently, an oligonucleotide STAT3 decoy (CS3D) was used to treat mice previously exposed to the tobacco carcinogen nitrosamine 4-(methylnitrosamino)-1-(3-pyridyl)-1-butanone (NNK), which contained a sequence of double-stranded STAT3 DNA response elements and interrupted the STAT3 signaling by binding to STAT3 dimers, preventing them from initiating transcription at the native DNA binding site of STAT3. CS3D successfully reduced oncogenic signaling in the airway epithelium by reducing the expression of STAT3 and its downstream genes and favored a lung microenvironment with alleviated immunosuppression by reducing pulmonary M2 macrophages and MDSCs cells [59].

siRNA are short, double-stranded RNA molecules which could silence the expression of target mRNAs through an RNA-induced silencing complex (RISC) with highly efficiency and specificity. siRNA-based therapy has gained much attention in cancer treatment, and many siRNA-based drugs have already entered clinical trials [58]. For lung cancer, many studies have demonstrated that using siRNA to suppress STAT3 expression could successfully repress cancer development, reverse drug resistance (chemotherapeutic drugs and TKIs), and improve immunotherapy by inhibiting proliferation, migration, and ROS production of cancer cells as well as decreasing the expression of PD-L1 on immune cells [60,61,62].

Double-strand miRNAs, which mimic naturally occurring miRNAs, could replenish the function of altered miRNAs by binding to the 3′-untranslated region (3′-UTR) of target mRNAs, leading to mRNAs degradation and suppression [58]. Recent studies showed that transfecting cells with miRNAs not only inhibited the proliferation of NSCLC cells and reversed cisplatin resistance but also activated CD8^+^ T cells in a STAT3/PD-L1-dependent manner by binding to STAT3 3′-UTR [21,63].

In addition, lncRNAs have been demonstrated to regulate various stages of gene expression, positively or negatively correlated with the clinical outcomes of NSCLC. Over- or underexpression of target lncRNA that regulate the phosphorylation of STAT3 significantly inhibited the proliferation of NSCLC cells and reversed the resistance of afatinib to NSCLC cells [64,65].

The specificity and effectiveness of therapeutic RNA molecules make them popular therapeutic agents for diseases such as lung cancer. However, the rapid degradation of RNA molecules during circulation in vivo has become a key issue in RNA therapeutics. The most widely used transfection reagents are cationic lipid (Lipo 2000) or polymer (polyethylenimine), which might induce high cytotoxicity. Therefore, designing safe and effective delivery methods to protect RNA molecules from degradation and improve in vivo transfection efficiency is very important in RNA-based cancer therapy.

### 3.5. Agents Regulating the STAT3 Upstream Gene

Since STAT3 activity was found to be associated with many signal pathways, targeting its upstream regulators may be an alternative strategy to inhibit STAT3. Recently, cyclin-dependent kinase (CDK) family members, which were supposed to regulate the cell cycle, were demonstrated to promote RNA synthesis in genetic processes by regulating the STAT3 signaling pathway [66]. Therefore, CDK inhibitors successfully reduced the expressions of pSTAT3 and transcriptional target genes such as cyclin B1 and IL-6, leading to apoptosis of lung squamous cell carcinoma (LUSC) cells and inhibition of tumor growth in patient-derived xenograft (PDX) models [66,67]. In addition, the DNA damage-induced apoptosis suppressor (DDIAS) has been reported to promote the progression of lung cancer through the regulation of the STAT3 pathway, and DDIAS inhibitors were found to suppress the activation of c-Jun NH(2)-terminal kinase (JNK) or interfere with DDIAS/STAT3 binding [68,69]. Other targets, such as Ataxia Telangiectasia Mutated (ATM), have been reported to upregulate PD-L1 expression via the JAK1, 2/STAT3 pathway, mediating the cisplatin-resistance in lung cancer cells [70]. Chen and colleagues found that the inhibition of ATM using specific ATM inhibitor CP466722 or siRNA reduced JAK/STAT3 signaling and PD-L1 expression, thus inhibiting the EMT and metastatic potential of cisplatin-resistant lung cancer cells [70]. A newly discovered heat shock protein 90 (Hsp90) inhibitor, NCT-80, bound directly to the C-terminal ATP-binding pocket of Hsp90, disrupted the interaction between Hsp90 and STAT3, and degraded the STAT3 protein, thereby reducing CSC-like phenotypes of NSCLC cells and their sublines with acquired resistance to anti-cancer drugs [71].

In addition, STAT3 is also negatively regulated by several regulators. Overexpression of miR-218 exhibited an anti-CSCs effect in lung cancer by directly targeting the 3’-UTR of mRNAs of the IL-6 receptor and JAK3 gene [22]. In addition, Herman et al. found that protein tyrosine phosphatase receptor-T (PTPRT) negatively regulated STAT3 function by dephosphorylating STAT3 (Tyr705), and the silence of PTPRT using siRNA in NSCLC led to the increased expression of pSTAT3 (Tyr705) and STAT3 target genes such as cyclin D1 and Bcl-XL [72].
pharmaceutics-14-02787-t001_Table 1Table 1Current strategies to target STAT3 in lung cancer.CategoryDrugMechanismTherapeutic EffectR&D StageNon-oncology drugsNiclosamide [28](FDA-approved antihelminthic drug)Blockage of p-STAT3 binding to the promoter of PD-L1, leading to downregulation of PD-L1An enhancement in PD-L1 antibody by niclosamide was observed in the inhibition of NSCLC growth in vitro and in vivoPreclinical research
HJC0152 [35](Antihelminthic drug, derivative of niclosamide)Inhibition of the phosphorylation of STAT3 at Tyr705 and alteration of metabolic pathwaysHJC0152 reduced the capacity of cells to scavenge free radicals, leading to the generation and accumulation of ROS and inducing apoptosisPreclinical research
Pyrvinium [36](FDA-approved antihelminthic drug)Suppression of STAT3 phosphorylation at tyrosine 705 and serine 727, leading to metabolic lethality in KRAS-mutant lung cancerPyrvinium triggered ROS release, depolarized the mitochondrial membrane potential, and suppressed aerobic glycolysis in KRAS-mutant lung cancer cellsPreclinical research
Halofuginone (HF) [37](Anticoccidials)Suppression of ERKand STAT3 phosphorylation, and increase in p38 phosphorylationHF reduced cancer cell viability and induced cell cycle arrest and apoptotic cell deathPhase I clinical trial (NCT00027677)
Acetaminophen (AAP) [38](FDA-approved antipyretic and analgesic drug)Suppression of STAT3 activity by directly binding to STAT3 with high affinityAAP exhibited anti-CSC effects (3D spheroid formation, self-renewal, and expression of CSC markers) both in vitro and in vivoPreclinical research
Dihydroartemisinin (DHA) [39](Antimalarial drug)Suppression of STAT3 phosphorylation, leading to the downregulation of Mcl-1 and Survivin DHA enhanced ABT-263-induced cytotoxicity via inhibiting STAT3 and triggered apoptosis in NSCLC cells harboring EGFR or RAS mutationPreclinical researchNatural compoundsCurcumin (CUR) and its derivative [40,41]Inhibition of phosphorylation of STAT3, JAK, p38, and JNK, as well as their downstream genes (VEGF, Bcl-xL, Cyclin D1) Curcumin significantly inhibited cell migration and tube formation in vitro and inhibited tumor growth and angiogenesis in vivoPhase I clinical trial for NSCLC (NCT02321293)
Cucurbitacin I (CuI) [42]Disrupting actin filaments that are physically associated with JAK2 and STAT3 in A549 cells and regulating their phosphorylation CuI dose-dependently inhibited the phosphorylation of STAT3 and enhanced the phosphorylation of STAT1 in lung adenocarcinoma A549 cellsPreclinical research
Raddeanin A [43]Inhibition of p-STAT3 and STAT3 expression and reduction of mitochondrial membrane potential by generating ROSRaddeanin A treatment had no obvious effect on 16HBE cells viability, but it inhibited the viability and proliferation of A549 and H1299 cellsPreclinical research
Ginkgolide C (GGC) [44]Inhibition of the phosphorylation of STAT3 and STAT3 upstream kinasesGGC exposure significantly reduced NSCLC tumour growth without significant adverse effects by reducing the level of p-STAT3 in mouse tissuesPreclinical research
Eupatolide [45]Suppression of the activation of STAT3 in NSCLC cellsEupatolide enhanced the anti-tumor activity of the cisplatin and 5-Fluoracil (5-FU) and suppressed tumor growth in vivoPreclinical research
Proscillaridin A (PSD-A) [46]Inhibition of both constitutive and inducible STAT3 activations and reduction of STAT3 DNA-binding activity by increasing SHP-1 expression, reducing Src phosphorylation, and binding PSD-A to STAT3 SH2 domainPSD-A induced oxidative stress, ER stress, and mitochondrial dysfunction, resulting in apoptosis in A549 cellsPreclinical research
Ethanol extract of Scutellaria baicalensis (ESB) [47]Inhibition of STAT3 phosphorylation and downregulation of the target gene expression in EGFR TKI-resistant lung cancer cellsESB reduced cell viability, suppressed colony formation, and induced cell cycle arrest and apoptosis in EGFR TKI-resistant lung cancer cellsPreclinical research
Pterostilbene [48]Pterostilbene plus Osimertinib reversed osimertinib-induced phosphorylation of STAT3, YAP1, and CUB domain containing protein-1 (CDCP1)The combination of osimertinib and pterostilbene showed synergistic anti-proliferative effects in all EGFR-mutation-positive NSCLC cell linesPreclinical researchSmall molecular inhibitorsW2014 [49]Occupation of sub-pockets of the SH2 domain to prevent the dimerization and transcriptional activity of STAT3W2014 exhibited potent anti-tumor activities and sensitized resistant NSCLC cells to gefitinib both in vitro and in vivoPreclinical research
BBI608 [51,52,54](FDA-approved for gastro esophageal junction and pancreatic cancer)Docking with the SH2 domain of STAT3 SH2 domain leads to disruption of STAT3 phosphorylation and dimerization BBI608 enhanced the anti-tumor effect of gefitinib on EGFR-mutated NSCLC cells and overcame cisplatin resistance in NSCLCPhase I, II and III clinical trials for NSCLC (NCT02347917, NCT02826161)
LL1 [53]Interference with the binding of the SH2 domain affects the phosphorylation of STAT3LL1 sensitized the resistance cells to gefitinib with little toxicity both in vitro and in vivoPreclinical research
ACHP [50]Interaction with the SH2 domain to inhibit the activation and nuclear localization of STAT3 and to suppress the activation of JAK1, JAK2, and SrcACHP blocked the proliferation activity of NSCLC, induced apoptosis, and reduced the expression of tumorigenic proteins (survivin, Bcl-2, Bcl-xl, and cyclin D1)Preclinical research
BP-1-102 [55]Combination with RAF inhibitor AZ628 markedly abrogated the activation of the MEK/ERK signaling pathway in KRAS-mutant lung cancer cellsThe combination of AZ628 and BP-1-102 showed a strong synergistic effect on KRAS-mutant cells and significantly induced cell apoptosis compared with a single inhibitorPreclinical research
YHO-1701 [56]Preventing phosphorylated tyrosine peptides from binding to the SH2 domain of STAT3STAT3 inhibition by YHO-1701 effectively suppressed the adaptive survival of ALK-rearranged lung cancer cells by enhancing ALK inhibition (alectinib)-induced apoptosisPreclinical research
MMPP [57]Binding to the hydroxyl residue of threonine 456 in the DNA-binding domain (DBD) of STAT3 inhibited its phosphorylation and DNA binding activityMMPP showed anti-tumor activity similar to or better than docetaxel or cisplatin by inducing G1-phase cell cycle arrest and apoptosisPreclinical research
CS3D [59]Interruption of STAT3 signaling by binding to STAT3 dimers, rendering them unable to initiate transcription at native STAT3 DNA binding sitesCS3D successfully reduced oncogenic signaling in the airway epithelium of mice exposed to the tobacco carcinogen and reduced immunosuppressionPreclinical researchTherapeutic RNA moleculesmiR-4500 [63]Direct target of the 3′UTR of STAT3 to suppress STAT3 expressionmiR-4500 suppressed the cell proliferation, migration, and invasion and promoted apoptosis of human NSCLC cell linesPreclinical research
miR-526b-3p [21]Direct target of the 3′UTR of STAT3 to suppress STAT3 expressionmiR-526b-3p reversed cisplatin resistance, suppressed metastasis, and activated CD8^+^ T cells in a STAT3/PD-L1-dependent mannerPreclinical research
LncRNA *HAR1A* [64]Regulation of the phosphorylation of STAT3LncRNA *HAR1A* inhibited the proliferation of NSCLC cells in vitro and in vivoPreclinical research
Knockdown of LncRNA BLACAT1 [65]Suppression of the phosphorylation of STAT3Knockdown of LncRNA BLACAT1 reversed the resistance of afatinib to NSCLC cellsPreclinical researchTarget upstream regulators of STAT3Acetyl-bufalin [66]Disrupting the formation of CDK9 and STAT3 complex, and reducing the expressions of P-STAT3 and transcribed target genesAcetyl-bufalin inhibited tumor growth in CDX and PDX models of NSCLCPreclinical research
Palbociclib [67](FDA-approved for breast cancer)Inhibiting IL-1β and IL-6 expression, and blocking Src/STAT3 signaling in an RB-independent mannerPalbociclib-induced apoptosis in LUSC cellsPhase I, II and III clinical trials for lung cancer (NCT01291017, NCT03170206, NCT02022982, NCT04870034)
DGG-100629 [68]Inhibition of STAT3 phosphorylation via the JNK/NFATc1/DDIAS pathwayDGG-100629 inhibited the growth of lung cancer cells and patient-derived gefitinib-resistant lung cancer cells expressing NFATc1 and DDIASPreclinical research
Miconazole [69](FDA-approved antifungal drug)Disruption of DDIAS/STAT3 interaction to inhibit the phosphorylation of STAT3 tyrosine Y705 and its target genesMiconazole inhibited the growth and migration of lung cancer cells both in vitro and in vivoPreclinical research
CP466722 [70]Suppression of JAK1/2, STAT3, and PD-L1 by inhibiting ATMCP466722 suppressed EMT and metastatic potential of cisplatin-resistant lung cancer cellsPreclinical research
NCT-80 [71]Disruption of the interaction between Hsp90 and STAT3 and degradation of STAT3 proteinNCT-80 reduced the viability, colony formation, migration, and CSC-like phenotypes of NSCLC cells and their sublines with acquired resistance to anti-cancer drugsPreclinical research
miR218 [22]Target the 3’-UTR of the mRNAs of the IL-6 receptor and JAK3miR-218 reduced cell proliferation, invasion, colony formation, and tumor sphere formation in vitro and inhibited tumor growth in vivoPreclinical research
siRNA PTPRT [72]Silencing PTPRT with siRNA in NSCLC resulted in increased pSTAT3Tyr705 and upregulation of STAT3 target genes/Preclinical research


## 4. Nanoparticle-Based Delivery of STAT3 Inhibitors in the Treatment of Lung Cancer

Although the above STAT3 inhibitors have shown good therapeutic potential in preclinical trials of cancers such as lung cancer, they still face problems such as poor solubility and bioavailability, low tumor targeting, off-target toxicity, easy degradation (therapeutic RNA molecules), and drug resistance during the process of clinical translation.

Nanoparticles (NPs) are generally believed to increase drug concentrations in cancer cells, protect RNA from degradation, and reduce toxicity to normal cells through passive (enhanced permeability and retention effect, EPR effect) or active targeting strategies. A variety of nanocarriers, including lipid nanoparticles (LNPs)/liposomes, inorganic NPs, polymeric micelles, and extracellular vesicles (EVs), have shown great potential in the diagnosis and treatment of a variety of cancers that rely primarily on the oncogenic STAT3 signaling pathway (Figure 2). However, the development of nanocarriers for lung cancer-targeted delivery of STAT3 inhibitors is quite challenging due to several hurdles presented by the lungs’ anatomy and pathology/physiology that need to be carefully addressed. The applications of NPs-mediated STAT3 inhibitors delivery in other cancers can be useful as guidance (Table 2). In addition, the challenges in lung cancer-specific drug delivery have also been summarized.

### 4.1. LNPs/Liposomes

LNPs, typically surrounded and stabilized by lipid bilayers with an aqueous (liposomes), oil, solid, or amorphous core (for nucleic acid delivery), are one of the most common nanoparticle formulations used to deliver anti-cancer drugs or genes because of their stable drug encapsulation and enhanced delivery efficiency [73]. The initial success of several liposome/LNPs-based drugs has fuelled further clinical investigations [74]. The improved therapeutic outcomes of Doxil^®^ (a liposome formulation of doxorubicin) and the recent approval by the U.S. Food and Drug Administration (FDA, Silver Spring, MD, USA) for LNP-loaded mRNA vaccines used to prevent COVID-19 have made LNPs a promising drug delivery system for various diseases [75]. Due to uniform particle size distribution, LNPs have exhibited superior biodistribution in vivo, which has established them as outstanding drug carriers for lung cancer treatment [76]. Moreover, since the main components of LNPs (e.g., phospholipids and cholesterol) are very similar to pulmonary surfactants in mammals, several highly biocompatible and biodegradable inhaled LNPs are promising candidates for pulmonary drug delivery [77]. Studies indicated that inhaled LNPs could increase the therapeutic effect of the drug and decrease the systemic toxicity simultaneously because they are able to restrict the drug effect in the pulmonary system for a prolonged time [78].

Villanueva and colleagues reported the potential of STAT3 decoy-loaded cationic lipid microbubbles (STAT3-MB) combined with ultrasound-targeted microbubble cavitation (UTMC) in the treatment of head and neck squamous cell carcinoma (HNSCC). The STAT3 decoy was loaded via charge–charge interaction. The formed STAT3-MB combined with UTMC treatment promoted the delivery of cell-impermeant oligonucleotides exclusively to sites exposed to the ultrasound beam, significantly inhibiting tumor growth and prolonging the survival in CAL33 tumor-bearing mice compared to the negative control groups, which was associated with the downregulation of the expression of target genes Bcl-xL and Cyclin-D1 at the RNA transcription and protein levels [79]. In order to achieve better tumor targeting ability and efficient cell transfection, surface modification of liposomes/LNPs by covalently conjugating specific ligands has gained much attention. A study reported that α5β1 integrin receptor selective liposomes prepared via conventional thin-film hydration method containing RGDK-lipopeptide simultaneously delivered a small molecule STAT3 inhibitor (WP1066) and STAT3 siRNA to brain tumors. It was found that WP1066/STAT3-siRNA-loaded liposomes were internalized in glioblastoma cells via integrin α5β1 receptors and selectively accumulated in brain tissues of glioblastoma-bearing mice, thus significantly improving the overall survival of orthotopically-established glioblastoma-bearing mice [80]. Solid lipid nanoparticles (SLNs) are another promising delivery system for small molecules and genes due to their good biocompatibility and physical stability. A recent study developed and evaluated the use of cationic SLNs for delivery of RNA interference (RNAi)-mediating plasmid DNA to downregulate STAT3 in cisplatin-resistant lung cancer cells. Cationic SLNs were prepared by a modified hot microemulsion method, and these cSLN:plasmid DNA complexes successfully encoded anti-STAT3 short hairpin RNA, reduced STAT3 expression, and improved the sensitivity of the cisplatin-resistant Calu1 cell line to cisplatin (Figure 3) [81].

Since STAT3 activation is associated with immune suppression, inhibition of STAT3 activation by different strategies has shown promising results in cancer immunotherapy. For example, ablation of STAT3 in mice could induce potent anti-tumor immunity by increasing the production of IL-12 and tumor necrosis factor α (TNFα), reducing the production of IL-10, and inducing M1-like reprogramming of murine macrophages [82]. To this end, Møller et al. reported a type of long-circulating liposomes (CA-LCL-αCD163), which were passively inserted lipidated CD163 (markers of M2-polarized macrophages) into the liposome lipid bilayer and packed with STAT3 inhibitor corosolic acid (CA). The CA-LCL-αCD163 liposomes were able to target macrophages with high CD163 expression, inhibit IL-6-induced STAT3 activation, and induce the production of pro-inflammatory cytokines, resulting in reprogramming tumor-associated macrophages (TAMs) from a tumor-supporting (M2-like) phenotype towards a tumoricidal (M1-like) phenotype [82]. Another study prepared doxorubicin-loaded, cholesterol-free CA liposomes (DOX/CALP) based on PEGylated liposomal doxorubicin (DOXIL^Ⓡ^) by replacing its cholesterol with CA. They found that DOX/CALP displayed higher in vitro cellular uptake and tumor spheroid permeation, as well as stronger anti-tumor cytotoxicity, compared to doxorubicin-loaded cholesterol liposomes (DOX/LP). In addition, the pSTAT3 level in the DOX/CALP group was significantly suppressed, and fewer intratumoral macrophages were observed in the DOX/CALP group, further suggesting that CALP as a functional delivery nanocarrier has some advantages over classic liposomes, and hence could enhance the efficacy of chemotherapeutic drugs [83]. In addition to STAT3 inhibitory drug delivery, targeting TAMs with STAT3 siRNA-loaded LNPs to modify their function responsible for M2 polarization could also be used to reverse the tumor-promoting function of TAMs. Harashima et al. fabricated a type of pH-sensitive LNPs (CL4H6-LNPs) for targeted delivery of STAT3 siRNA to TAMs. The silencing of STAT3 and hypoxia-inducing factor 1α (HIF-1α) led to an increase in levels of infiltrated macrophages (CD11b^+^ cells) and M1 macrophages (CD169^+^ cells) in the tumor microenvironment (TME), achieving novel macrophage-based cancer immunotherapy [84].

Studies also revealed that the activation of STAT3 was strongly associated with the expression of PD-L1 in multiple cancers, and inhibition of STAT3 can reduce the expression of PD-L1, resulting in the improved therapeutic effect of checkpoint inhibitors [26]. Li et al. synthesized a novel IL-20 receptor subunit alpha (IL20RA)-targeted liposomal NP that encapsulates the STAT3 inhibitor stattic (NP-Stattic-IL20RA) to inhibit breast cancer. They demonstrated that IL20RA could promote the stemness of breast cancer cells via the JAK1-STAT3-SOX2 signaling pathway and regulate the expression of PD-L1 to modulate the immune microenvironment. NPs-Stattic-IL20RA combined with anti-PD-L1 antibody effectively inhibited the stemness of cancer cells and improved the tumor immune microenvironment, resulting in an increase in the efficacy of chemotherapy [85]. Recently, a pH-responsive liposome (Liposome-PEO, LP) loaded with apatinib (AP) and cinobufagin (CS-1) and coated with a hybrid membrane (R/C) (LP-R/C@AC NPs) was prepared for combined treatment of gastric cancer. LP-R/C@AC efficiently killed tumor cells by inhibiting the vascular endothelial growth factor receptor 2 (VEGFR2)/STAT3 pathway and reverse tumor immunosuppression by inhibiting the expression of PD-L1 and matrix metalloproteinase 9 (MMP-9), showing the dual advantages of targeting tumor cells and immune escape [86].

### 4.2. Inorganic Nanoparticles

Inorganic NPs, including gold/silver NPs, mesoporous silica NPs, and magnetic NPs, have been extensively explored in cancer theranostics over the past two decades due to their advantages of facile preparation, excellent biocompatibility, and wide surface conjugation chemistry [87]. In addition, these inorganic NPs, including gold NPs and magnetic NPs with minimal toxicity, good stability, and powerful imaging properties, are widely used in lung cancer diagnosis, acting as nanoprobes in computed tomography (CT) or magnetic resonance imaging (MRI) for molecular imaging of lung cancer in the clinic [88]. However, the potential cellular toxicity and adverse effect of magnetic NPs should not be ignored; thus, the size, concentration, and exposure time must be carefully understood [89].

Gold NPs are widely used for drug delivery because of their easy synthesis, high surface volume, and functionalization [90]. A study reported the synthesis of curcumin-loaded or curcumin/paclitaxel co-loaded gold NPs for the treatment of triple-negative breast cancer [90]. The results demonstrated that gold NPs loaded with curcumin with/without paclitaxel exhibited anti-cancer and anti-metastatic properties by downregulating the expression of STAT3 and downstream genes (MMPs, VEGF, and Cyclin D) [90]. Another study reported the development of layer-by-layer assembled gold NPs (LbL-AuNP) containing anti-STAT3 siRNA and imatinib mesylate (IM) to treat melanoma. Notably, LbL-AuNP prepared using sequential adsorption of natural polyelectrolytes, chitosan, and sodium alginate resulted in a positive charged surface, which could be utilized for iontophoresis therapy to enhance skin penetration in the local treatment of melanoma at an early stage. The topical iontophoretic application of dual-drug loaded LbL-AuNP significantly inhibited tumor growth and STAT3 expression in mouse melanoma models, compared with the control treatments [91].

Iron oxide is another commonly used material for the synthesis of inorganic NPs. Superparamagnetic iron oxide nanoparticles (SPIONs) composed of superparamagnetic magnetite (Fe_3_O_4_) or maghemite (Fe_2_O_3_) at certain sizes are considered to be highly efficient nanocarriers for anti-cancer therapeutics [92]. However, non-specific binding to serum proteins and rapid clearance from the bloodstream are major challenges for the application of SPIONs. Niaragh and colleagues coated SPIONs with positively charged chitosan derivatives such as trimethyl chitosan (TMC) and thiolated chitosan (ChT) to improve the stability of SPIONs and siRNA loading potential. In addition, Hyaluronate (HA) and TAT peptide were conjugated on the surface of SPIONs to facilitate their tumor tissue penetration and tumor cellular uptake. These HA-conjugated TAT-chitosan-SPION (SPION-TMC-ChT-TAT-H) NPs successfully co-delivered STAT3/HIF-1α siRNA, and significantly inhibited STAT3/HIF-1α gene-driven tumor proliferation, migration, and metastasis [93].

In addition to metal NPs, inorganic, nonmetallic materials have attracted considerable attention in the field of drug delivery. For example, silica NPs have been successfully used for gene and drug delivery, owing to their ability to improve the stability of protected substances in their cores without interfering with their chemical and physical properties [94]. One study developed a type of SiO_2_ NPs (ZnAs@SiO_2_), which encapsulated arsenic trioxide (ATO) by a “one-pot” reverse emulsification approach (Figure 4) [95]. The ZnAs@SiO2 NPs reduced the expression of stemness markers (CD133, Sox-2, and Oct-4) and EMT markers (E-cadherin, Vimentin, and Slug) by inhibiting the STAT3 signaling pathway and thus inhibited tumor spheroid formation in vitro and tumor initiation and metastasis in vivo [95]. Calcium phosphate NPs (CaP) are also utilized for gene delivery with negligible cytotoxicity and superior biodegradability. Furthermore, CaP dissolves in acidic endosomes and helps the cargo be released into the cytosol through the endosome rupture [96]. Li et al. proposed a novel hybrid vesicle with inorganic CaP as the kernel and with HA modification on the surface (CaP@HA), for targeted delivery of STAT3-decoy ODNs. They demonstrated that the STAT3-decoy ODNs-loaded CaP@HA vesicles effectively suppressed the expression of STAT3 and its downstream target gene mucin 4 (MUC4), which could interfere with the interaction of Trastuzumab (TRAZ) and human epidermal growth factor receptor 2 (HER2), thereby efficiently reversing TRAZ resistance in anti-HER2 therapy [96]. Similarly, Ke et al. proposed another inorganic kernel of CaP as the core of reconstituted low-density lipoprotein (LDL) nanovehicles (CaP@LDL) for targeted delivery of STAT3-decoy ODNs to reverse the resistance of tumor necrosis factor-related apoptosis-inducing ligand (TRAIL). The results showed that CaP@LDL nanovehicles possessed LDL-mimicking pharmacokinetics, which enabled them to efficiently deliver STAT3 decoy-ODNs to overcome TRAIL resistance by blocking the expression of STAT3 and downstream anti-apoptotic target genes (Bcl-2, Bcl-xl, and Mcl-1) [97].

### 4.3. Polymeric Micelles

Polymeric micelles with core–shell structures have been extensively applied for the delivery of small molecules, therapeutic genes, antibodies (Abs), and RNA-based therapeutics due to their stable structure, good biocompatibility, high drug loading, outstanding pharmacokinetics, and preferential tumor accumulation [98]. Despite the passive targeting ability of polymeric micelles under the EPR effect, facile methods were used to modify their surfaces with specifically targeting ligands, which could target overexpressed receptors on lung cancer cells (e.g., EGFR and CD44 receptors) in order to achieve improved tumor-specific targeting [99,100]. Moreover, these targeting ligands could partially inhibit the function of overexpressed receptors and regulate over-activated pathways, thus improving drug action in tumor-specific lung tissues [99].

Micellar formulations containing cationic polymers, such as poly-L-lysine (PLL) and poly(ethylenimine) (PEI), were able to deliver siRNA via electrostatic interaction, thus reducing RNA degradation and enhancing intracellular accumulation [101]. For example, researchers developed cholesterol-modified dicer-substrate siRNA (Chol-DsiRNA) Polyplexes, which were formed by the encapsulation of STAT3 siRNA with 50 poly-L-lysine residues and 5 kDa polyethylene glycol. Chol-DsiRNA Polyplexes demonstrated improved anti-tumor efficacy with good tolerance by efficiently inhibiting STAT3 [102]. Another study prepared the dual-targeting system by electronic self-assembly, which was composed of folic acid-conjugated carboxymethyl chitosan for targeting and cationic chitosan derivatives for STAT3 siRNA package (FA-OCMCS/ N-2-HACC/siSTAT3). The NPs dramatically reduced STAT3 expression in M2 macrophages and Lewis lung cancer cells and shifted the phenotype of macrophages from M2 to M1, resulting in the suppression of tumor growth [103].

The unique core–shell structure of micelles is able to co-deliver two or more therapeutic agents through self-assembly, which is considered an effective combination therapy strategy to overcome drug resistance, tumor metastasis, and immunosuppression. A study developed a micellar delivery system (PEG-PLA NPs) based on FDA-approved poly(ethylene glycol) (PEG)-poly(lactic acid) (PLA) for the co-delivery of Erlotinib (ELTN, EGFR-TKI) and fedratinib (FDTN, JAK2 inhibitor). A synergistic anti-cancer effect was achieved by PEG-PLA NPs in ELTN-resistant NSCLC by downregulating the expression levels of proteins in the JAK2/STAT3 signaling pathway, including pEGFR, pJAK2, pSTAT3, and survivin [104]. PEG-PLA was also utilized to co-deliver gefitinib (Gef) and Cyclosporin A (CsA), and the results showed that CsA formulated in NPs sensitized Gef-resistant NSCLC to Gef treatment by inactivating the STAT3/Bcl-2 signaling pathway (Figure 5) [105]. Li and colleagues developed a multifunctional nanocomplex to simultaneously deliver paclitaxel (PTX) and STAT3 siRNA (siSTAT3) to inhibit tumor growth and prevent metastasis. PTX and siSTAT3 were encapsulated into the synthesized polyethyleneimine-polylactic acid-lipoic acid (PPL) micelles through hydrophobic or electrostatic interaction, respectively. Furthermore, the negatively charged HA was coated on the surface of the drug-loaded nanocomplex (HA/^siSTAT3^PPL_PTX_) in order to effectively enter CD44-overexpressed 4T1 cells via an active targeting mechanism. HA/^siSTAT3^PPL_PTX_ exhibited superior anti-tumor efficacy and effectively reduced the lung metastasis of 4T1 cells by silencing the expression of STAT3 and pSTAT3 [106]. Another study reported the development of a pulmonary delivery system (FM@PFC/siRNA) based on perfluorocarbon (PFC) nanoemulsions for co-delivery of C-X-C motif chemokine receptor 4 antagonist (FM) and anti-STAT3 siRNA. The FM@PFC/siRNA nanoemulsions inhibited both CXCR4 and STAT3 signaling, induced apoptosis and anti-invasive activity, and overcame the immunosuppressive TME, achieving good efficacy in lung metastatic tumor models [107].

Studies have shown that siRNA delivery by polymeric micelles though electrostatic interaction is not very stable, resulting in inefficient transfection efficiency and high variability [108]. In addition, cationic NPs are toxic and potentially capable of inducing immunogenicity in animals and humans. Furthermore, siRNAs are susceptible to being cleared by the kidneys because excess cationic components could make NPs easy to disassemble at the glomerular basement membrane [109]. To solve these problems, researchers used dithiothreitol (DTT) to reduce the disulfide-protecting groups of siRNA at the 3′ end of the sense strand, and the obtained siRNA was reacted with Pluronic F108 functionalized with pyridyl disulfide groups. This covalent conjugation of siRNA with Pluronic F108 provides a stable nanoparticle formulation with efficient siRNA loading, achieving consistent target-specific gene knockdown [108]. Shi et al. reported a new class of cation-free polymeric micellar spherical nucleic acid (SNA), which can deliver both STAT3 siRNA and temozolomide (TMZ) in a controlled release manner. The siRNA-disulfide-poly (N-isopropylacrylamide) (siRNA-SS-PNIPAM) diblock copolymer could self-assemble to form SNAs, cross the blood–brain barrier, and enter brain tumor cells through a scavenger receptor-mediated mechanism, achieving a remarkable synergistic effect against TMZ-resistant tumors [109].

Micelles for targeted STAT3 delivery may potentially modulate immunosuppressive TME, leading to improvements in immunotherapy. Jiang and colleagues synthesized a pH-responsive copolymer PEG-poly(lysine-DMMA)-poly(phenylalanine) to co-encapsulate two prodrugs, Gemcitabine-C18 and NI-HJC0152 (STAT3 inhibitor). NI-HJC0152 responded to the progressively intensive hypoxia in tumor tissue to yield parental HJC0152 that inhibits STAT3, leading to the reversal of tumor immunosuppression through modulating TAM polarization, recruiting cytotoxic T lymphocytes, and reducing regulatory T cells. In addition, inhibition of STAT3 also downregulated the expression of cytidine deaminase (CDA) and α-smooth muscle actin (α-SMA), thus relieving the resistance of gemcitabine [110]. Another study constructed a self-assembling vehicle-free multi-component anti-tumor nanovaccine (SVMAV) using an unsaturated fatty acid docosahexaenoic acid (DHA)-conjugated antigen and R848 (a Toll-like receptor 7/8 agonist) to encapsulate stattic (STAT3 inhibitor). The obtained SVMAV efficiently migrated into lymph nodes and primed CD8^+^ T cells to exert neoantigen-specific killing by promoting antigen uptake of dendritic cells (DCs), stimulating DCs maturation, and enhancing antigen cross-presentation, and finally achieved a robust anti-tumor effect in primary and lung metastasis models of melanoma [111].

### 4.4. Extracellular Vesicles

Studies have demonstrated that multiple cell types can excrete extracellular vesicles (EVs) with phospholipid bilayer membrane-bound structures, and EVs have shown great potential as drug delivery vehicles due to their nano-sized structure and ability to transport bioactive cargos between cells or tissues [112]. According to formation mechanism and typical size, Evs are mainly classified into three categories: exosomes (30–150 nm), microvesicles (MVs), or microparticles (MPs) (100–1000 nm) and apoptotic bodies (500–2000 nm) [113]. Compared with other nanocarriers, EVs have some advantages in lung cancer treatment, such as low toxicity, low immunogenicity, the ability to cross biological barriers, and the realization of multifunction through chemical or genetic modifications [112]. For example, exosomes modified with CD47 protein on the surface were able to evade phagocytosis by macrophages, leading to prolonged circulation time [114]. Furthermore, exosomes delivered from brain endothelial cells had the ability to cross the blood-brain barrier, exhibiting the potential ability to treat lung cancer with brain metastasis [115]. However, the acquirement of abundant exosomes with high quality is quite costly, and the lack of standardized methods to isolate, purify, and store exosomes has limited large-scale production and clinic translation [116]. Therefore, there is an urgent need to establish standard protocols to ensure the mass and consistent production of exosomes.

EVs from various origins hold great promise in cell-free anti-cancer treatment. For example, mesenchymal stem cell (MSCs)-derived EVs have shown unique advantages as carriers for anti-cancer drugs due to their lower immunogenicity and tumor migration capacity [117]. Qian et al. prepared EVs from human umbilical cord MSCs (huc-MSCs) transfected with adenovirus encoding Lipocalin-type prostaglandin D2 synthase (L-PGDS). EVs-L-PGDS inhibited the phosphorylation of STAT3 and the expression of downstream stem cell markers (Oct4, Nanog, and SOX2), thus inhibiting in vitro cancer cell proliferation and in vivo tumor growth [117]. Neural stem cell (NSCs)-derived exosomes have also been reported as vehicles for delivering oligonucleotide therapeutics (CPG-STAT3 antisense oligonucleotides, CpG-STAT3 ASO) to the glioma microenvironment, as NSCs have been shown to traffic into hypoxic areas of gliomas and secondary brain metastases. The results demonstrated that CpG-STAT3 ASO encapsulated NSCs/EV significantly activated glioma-associated myeloid cells and inhibited tumor progression in mice [118]. In addition, plant-derived nanovesicles for drug delivery have been discussed in recent years due to their safe and cost-efficient characteristics. Chen et al. obtained cucumber-derived nanovesicles (CsDNVs) at high yield and low cost, which may be natural nanocarriers that contain Cucurbitacin B (CuB, STAT3 inhibitor). They demonstrated that these CsDNVs enhanced the anti-cancer effects of CuB by improving its bioavailability [119]. Notably, owing to the remarkable capability for penetration of the BBB, exosomes might improve the prognosis of glioblastoma (GBM). Ye and colleagues prepared Angiopep-2 (An2)-conjugated (STAT3) siRNA-loaded exosomes (Exo-An2-siRNA) derived from human M1 macrophages. Exo-An2-siRNA could boost BBB permeation and GBM targeting by exploiting the tumor-homing characteristic of M1 macrophages and specifically targeting (low-density lipoprotein receptor-related protein 1) LRP-1 ligands at the surfaces of both GBM cells and BBB endothelial cells, resulting in the favorable inhibition of the proliferation of orthotopic U87MG xenografts (Figure 6) [120].

### 4.5. Challenges in Lung Cancer-Targeted Drug Delivery

Although intravenous injection of nanomedicine is commonly used in other cancers, inhalation of nano-delivery systems is an additional administration route for lung cancer due to the unique anatomical and physiological characteristics of the lungs. However, due to the complex molecular and biochemical composition of the lung tissue, different biological barriers to drug delivery in lung cancer should be taken into consideration when designing efficient strategies for nanotechnology in lung cancer.

The presence of mucus in the respiratory system is the key mechanical barrier of the pulmonary region [121]. Mucins could form complex mesh by interacting with other mucin molecules and glycans in the mucins could provide negative charge, allowing NPs of different sizes and positive charges to be deposited in the mucus layer [121,122,123]. The pulmonary surfactant is one of the crucial chemical barriers for NPs to overcome before reaching the pneumocytes. Studies indicate that proteins in pulmonary surfactants prefer to bind magnetic NPs [124]. Moreover, hydrophilic NPs are easily trapped by the surfactant layer [125]. Proteolytic enzymes (e.g., cathepsin H) are another chemical barrier as they are responsible for the hydrolysis of protein and peptides of the NPs [126,127].

In addition, the complex tumor microenvironment (TME) is another biological barrier in the drug delivery of lung cancer. One of the most critical stromal cells in TME of lung cancer, namely cancer-associated fibroblasts (CAFs), has been demonstrated to play a key role in remodeling the tumor stroma and increasing the stiffness of the extracellular matrix (ECM), which might restrict the diffusional movement of NPs in tumor cells [128,129]. Macrophages in the TME, part of the clearance system, were found to be able to engulf, degrade, and remove NPs, thus affecting the number of NPs entering the tumor site [121]. Furthermore, the engulfment of NPs might induce the release of pro-inflammatory cytokines by macrophages, resulting in the activation of the immune system [130].
pharmaceutics-14-02787-t002_Table 2Table 2Nanoparticles for the targeted delivery of STAT3 inhibitors in cancer treatment.NameNanoparticles TypeExcipientsSTAT3 InhibitorManufacturing MethodsSize and ζ PotentialLE (%)EE (%)ModificationApplication in Cancer TreatmentSTAT3-MB [79]Cationic lipid microbubblesDSPC, DSEPC, DSPG and PEG-40STAT3 decoy oligonucleotideDSPC, DSEPC, DSPG, and PEG-40 were dissolved at a molar ratio of 100:43:1:4.5 in chloroform and dried. The dried lipid film was hydrated using isotonic saline and sonicated. The STAT3 decoy was carried via charge–charge interaction.2.45 ± 0.35 μm10 μg of STAT3 decoy per 1 × 109 microbubbles//STAT3-MB, in conjunction with UTMC, inhibited tumor growth in HNSCC.WP1066 + STAT3 siRNA-RGDK-lipopeptide [80]LiposomeDOPC, cholesterol, DSPE-PEG(2000) AmineSTAT3 inhibitor WP1066 and STAT3 siRNARGDK tetrapeptides were first synthesized, and thentetrapeptides were coupled with N,N-di-n-hexadecyl-N-2-aminoethylamine. The thin-film hydration method was followed to prepare the liposome.138.5 ± 6.4 nm; 8.7 ± 0.4 mV/96–98% for WP106 and 91% ± 3.2 for siRNAModified with RGDK-lipopeptideThe liposome was internalized in glioblastoma cells via integrin α5β1 receptors and accumulated in the brain tissue of glioblastoma-bearing mice, resulting in enhanced anti-cancer efficacy.cSLN:plasmid DNA complexes [81]Solid lipid nanoparticlesPATO5, C888, CTAB, DDAB and EQ1plasmid DNACationic SLNs were prepared by a modified hot microemulsion method, and CTAB, DDAB, or EQ1 were added to provide a positive charge.92.6 and 97.8 nm; 10.5 and 8.9 mV ///These cSLN:plasmid DNA complexes improved the sensitivity of cisplatin-resistant cells to cisplatin by encoding anti-STAT3 short hairpin RNA.CA-LCL-αCD163 [82]LiposomeHSPC, cholesterol and mPEG_2000_-DSPESTAT3 inhibitor corosolic acidHSPC, cholesterol, and mPEG_2000_-DSPE were dissolved at a molar ratio of 55:40:5 in methanol and dialyzed. The lipids were then incubated with CA for 60 min at 65 °C. The lipidated anti-human CD163 antibody was inserted into the CA-LCLs.62.0 ± 5.4 nm//Modified with CD163CA-LCL-αCD163 may target macrophages with high CD163 expression and reprogram TAMs from an M2-like phenotype towards an M1-like phenotype.DOX/CALP [83]LiposomeHSPC, cholesterol and mPEG_2000_-DSPEChemotherapeutic drug DOXHSPC, cholesterol, and mPEG_2000_-DSPE were mixed at a molar ratio of 56.3: 38.4: 5.3. CALP was prepared by thin film hydration methods via replacing cholesterol in cholesterol liposomes. The doxorubicin HCl was mixed with CALP for 30 min.97.00 ± 3.50 nm; −21.3 ± 2.46 mV10.76% ± 0.1798.39% ± 0.51Cholesterol in the liposome was replaced by corosolic acidDOX/CALP displayed strong cytotoxicity, suppressed intratumoral macrophages, and inhibited tumor growth in vivo. CL4H6-LNPs [84]Lipid nanoparticlescationic lipid CL4H6, cholesterol, DMG-PEG_2000_ and DSG-PEG_2000_STAT3 siRNA and HIF-1α siRNAThe LNPs were prepared by alcohol dilution procedure at a molar ratio of 60/40/1 mol% of CL4H6, cholesterol and PEG-lipid. siRNA was mixed with LNPs via the electrostatic interaction under acidic conditions.94.09 ± 3.10 nm; 0.21 ± 0.63 mV/92.50% ± 1.25/CL4H6-LNPs increased the levels of infiltrated macrophages (CD11b+ cells) into the TME and had a tendency to increase the concentration of M1 macrophages (CD169+ cells).NP-Stattic-IL20RA [85]LiposomeDOPE, DOPC and cholesterolSTAT3 inhibitor StatticDOPE, DOPC and cholesterol were mixed at a molar ratio of 1:1:1 to form liposomes. IL20RA-PEG conjugates (1:100 molar ratio) and Stattic were then mixed with liposomes.120.4 ± 2.91 nm; −11.3 ± 0.092 mV//Modified with monoclonal anti-IL20RA antibodyNP-Stattic-IL20RA targeted IL20RA^+^ tumor cells to regulate the expression of PD-L1 via the JAK1-STAT3-SOX2 signaling pathway, inhibit the stemness, and modulate the immune microenvironment.LP-R/C@AC NPs [86]LiposomeLecithin, cholesterol, DSPE-PEOz,STAT3 inhibitor Cinobufagin and VEGFR-2 TKIs ApatinibThe drug-loaded liposome was formed via the thin-film hydration method (Lecithin, cholesterol, DSPEPEOz, AP, and CS-1 at a molar ratio of 2: 1: 0.4: 2: 0.006–0.05) and was mixed with hybridmembrane (R/C) for 3 h.108 nm;−7.5 mV/94.2% for Apatiniband 99.9% for CinobufaginCoated with red blood cell membrane and cancer cell membraneLP-R/C@AC NPs efficiently killed tumor cells and reversed tumor immunosuppression by inhibiting the VEGFR2/STAT3 signal pathway.Au-CP [90]Gold nanoparticleHAuCl_4_STAT3 inhibitor Curcumin alone or combined with chemotherapeutic drug PaclitaxelHAuCl_4_ (200 μL, 10−2 M) solution was added to a 4.8 mL water fraction of Curcumin and Paclitaxel, and Au-CP was purified by ultracentrifugation.128 ± 10 nm; −3.0 ± 1.1///Au-CP demonstrated an excellent synergistic effect on breast cancer by downregulating the expression of STAT3 and its targeted genes both in vitro and in vivo.LbL-AuNP [91]Gold nanoparticleHAuCl_4_·3H_2_OSTAT3 siRNA and TKI imatinib mesylateAuNP-CS/siRNA particles were incubated with a chitosan solution containing IM and purified by centrifugation.197.8 ± 18.7 nm; 46.8 ± 2.7 mV19.2% ± 3.058.3% ± 7.5/LbL-AuNP could be utilized for iontophoresis therapy to enhance the skin penetration and tumor inhibition of melanoma at an early stage.SPION-TMC-ChT-TAT-H [93]Iron oxide nanoparticles FeCl_3_·6H_2_O, FeCl_2_·4H_2_O,chitosan, hyaluronate and TAT peptideSTAT3 siRNA and HIF-1α siRNAFeCl_3_·6H_2_O and FeCl_2_·4H_2_O were mixed to form SPION, and then TMC and ChT were coated on SPION via electrostatic reactions. TAT peptide and hyaluronate were subsequently added. Finally, siRNA was incubated with the mixture.118 ± 4 nm; 20 ± 1 mV//Modified with HA and TAT peptideThe NPs targeted and penetrated CD44^+^ cancer cells, inhibiting STAT3/HIF-1α gene-driven proliferation, migration and metastasis of cancer.ZnAs@SiO_2_ [95]Silica nanoparticlesZnCl_2_, disodium silicate, TEOS and SiO_2_STAT3 inhibitor Arsenic trioxide Zinc arsenite (ZnAsOx) NPs were synthesized by a reverse microemulsion method, and ZnAsOx NPs were encapsulated in the SiO_2_ matrix.51 ± 3 nm ///ZnAs@SiO2 NPs inhibited the formation of tumor spheroids and tumorigenesis and downregulated the expression of stemness and EMT markers by inhibiting the STAT3 signaling pathway.CaP@HA [96]Calcium phosphate nanoparticlesDOHAP, CaCl_2_STAT3-decoy ODNsDOHAP were synthesized by conjugating DOAP to the carboxyl groups of HA and CaP were synthesized via an inverse microemulsion system. DOHAP and CaP were mixed, centrifuged and purified.51.0 ± 3.4 nm; −18.6 ± 1.9 mV/78%Modified with HACaP@HA effectively suppressed the expression of STAT3 and MUC4, significantly eliminated the interaction between MUC4 with TRAZ and HER2 receptors and efficiently reversed TRAZ resistance in anti-HER2 therapy.CaP@LDL [97]Calcium phosphate nanoparticlesCaCl_2_, low-density lipoprotein, DOPC and cholesterolSTAT3-decoy ODNsCaP were synthesized via an inverse microemulsion system, and low-density lipoprotein was isolated from the plasma of healthy donors. CaP cores were added to the LDL-sucrose residue, together with DOPC and cholesterol, at a molar ratio of 1:1.40–45 nm; −4.2 ± 2.1 mV/75%Modified with low-density lipoproteinCaP@LDL regulated TRAIL resistance by blocking STAT3 signaling and the expression of downstream anti-apoptotic genes.Chol-DsiRNA Polyplexes [102]Polymeric micellesPEG [5K], PLL [30]STAT3 siRNAAdd the 2X Chol-DsiRNA Complexation Solution dropwise to the 2X Polymer Complexation Solution at 1/1 (*v*/*v*), mixing the solution by pipette aspiration/dispensation (30 s) and incubating (RT, 30 min).////Chol-DsiRNA Polyplexes were therapeutically active against primary murine syngeneic breast tumors.FA-OCMCS/N-2-HACC/si-STAT3 [103]Polymeric micellesFolic acid, OCMCS and N-2-HACC.STAT3 siRNAN-2-HACC/siRNA nanocomplexes (N-2-HACC: siRNA = 10:1, *w/w*) were prepared, and the complex solution was further mixed with FA-OCMCS (FA-OCMCS/N-2-HACC/siRNA = 15:10:1, *w/w/w*).179.8 ± 7.2 nm; −21.56 ± 2.75 mV//Modified with folic acidThis dual-targeting system shifted M2 macrophages to M1 macrophages and inhibited tumor growth by reducing STAT3 expression in cancer cells and M2 macrophages.ELTN+ FDTN@PEG-PLA [104]Polymeric micellesPEG-OH, DGPEFDTN (JAK2 inhibitor) and ELTN (EGFR-TKI)PEG-PLA solution was firstly added to DGPE at a polymer/lipid molar ratio of 95:5. ELTN and FDTN solution (20 μL) were then added. The mixture was added dropwise to a citrate buffer solution and dialysed against PBS.∼120 nm12.2% ± 2.1 for ELTN and 4.2% ± 1.3 for FDTN//PEG-PLA NPs exhibited synergistic anti-cancer effects in ELTN-resistant NSCLC by inhibiting the JAK-STAT3 signal pathway.CSA/Gef–NPs [105]Polymeric micellesPEG_5k_-b-PLA_8k_Cyclosporin A (CsA) and gefitinib (Gef, EGFR-TKI)Polymeric NPs were prepared via the nanoprecipitation method. In total, 1 mg of CsA and 40 mg of PEG-PLA were mixed and added dropwise into DI water then the solution was evaporated and concentrated.37.1 ± 13.1 nm; 8.160 ± 9.381 mV///CsA formulated in NPs sensitized Gef-resistant cells and Gef-resistant tumors to Gef treatment by inactivating the STAT3/Bcl-2 signaling pathway.HA/^siSTAT3^ PPL_PTX_ [106]Polymeric micellesPLA, LA and PEISTAT3 siRNA and chemotherapeutic drug PaclitaxelAfter activating the carboxyl group of PLA-LA by HOBt, and EDC, PEI was added dropwise, and the reaction mixture was dialyzed. PTX and siSTAT3 were added dropwise to the micelle solution, and HA was mixed subsequently.195 nm; −23 mV8.9%98.7%Modified with HAHA/^siSTAT3^PPL_PTX_ exhibited anti-tumor efficacy against breast cancer and effectively reduced invasion ability and lung metastasis by silencing the expression of STAT3 and p-STAT3.FM@PFC/siRNA [107]Polymeric micellesHMBA, AMD3100, HFBASTAT3 siRNA and CXCR4 antagonist FMFM was synthesized by Michael addition of an equimolar ratio of HMBA and AMD3100 and mixed with HFBA. The mixture was added with PFC and siRNA solution.∼190 nm; 28 mV///FM@PFC/siRNA induced apoptosis, anti-angiogenic effect, anti-invasive activity, overcame the immunosuppressive TME, and significantly inhibited lung metastasis.P-SS-STAT3/Ca [108]Polymeric micellesPluronic F108STAT3 siRNAThe COOP peptide having N-terminal cysteine (CACGLSGLGVA) was conjugated to the Pluronic surface by thiol-exchange reaction utilizing the pyridyl disulfide group in Pluronic F108343.5 nm; −12.6 mV to −0.153 mV///P-SS-STAT3/Ca enhanced the efficiency of STAT3 siRNA transfection, which was successfully demonstrated in cancer cells.siRNA-SS-PNIPAM [109]Polymeric micellesNIPAM monomer, RAFT reagent and azide groupmodified siRNASTAT3 siRNA and chemotherapeutic drug TMZThe siRNA-SS-PNIPAM polymers were synthesized by click chemistry via freeze−thaw treatment.40.9 nm11.33%51.12%/The cation-free polymeric micelles passed through the BBB and entered brain tumor cells via SRs mediated mechanism and achieved a synergistic effect against TMZ-resistant tumors.DMMA-pro drugs [110]Polymeric micellesNitroimidazole, stearyl chloride and DMMASTAT3 inhibitor NI-HJC0152 and chemotherapeutic drug GEM-C18DMMA were obtained via modifying the side chain of mPEG_5000_-poly(Lys)-poly(Phe).around 30 nm4.54% ± 0.33 for HJC0152 and 1.23% ± 0.99 for GEM-C1882.3% ± 1.2 for HJC0152 and 90.1% ± 1.3 for GEM-C18/The micelles reversed the immunosuppression of PDAC and relieved the resistance to GEM, resulting in a synergistic effect.SVMAV [111]Polymeric micellespeptide-CSSVVR-DHASTAT3 inhibitor static and TLR receptor 7/8 agonist R848Peptide-CSSVVR-DHA and R848-SS-DHA were synthesized, respectively, and they were mixed with static at a ratio of 1: 1: 1.~100 nm; 40.83 ± 1.52  mV///SVMAV efficiently stimulated DC maturation, enhanced antigen cross-presentation, and had an anti-tumor effect on primary and metastasis melanoma.EVs-L-PGDS [117]ExosomeHuman Umbilical Cord MSCsEnzyme L-PGDSAdenovirus encoding L-PGDS (Ad-L-PGDS) and vector (Ad-Vector) were added to the MSC medium; then the conditioned medium was centrifuged and concentrated for EV isolation.approximately 100 nm///EVs-L-PGDS could inhibit the phosphorylation of STAT3 and the expression of downstream stem cell markers, resulting in an efficient anti-cancer effect on gastric cancer both in vitro and in vivo.NSC (STAT3-CPG ASO) [118]ExosomeCpG-STAT3ASO conjugates and HB1.F3 NSCsSTAT3 ASO and CPG ASONSCs were incubated with CpG-STAT3ASO, and then culture media were collected, centrifuged at 2000× *g*, filtered through a 0.22 μm filter, followed by EV isolation using ultracentrifugation standard method.range of 40–200 nm///NSC entered the hypoxic areas of glioma and secondary brain metastases, activated glioma-associated myeloid cells and inhibited tumor progression in mice.CDNVs [119]NanovesiclesCucumbers/CDNVs were isolated by centrifuging juice from cucumbers (CsDNVs from sarcocarp and CpDNVs from pericarp) at 100,000× *g* for 60 min, followed by a typical exosome isolation protocol.180 nm and 129 nm; −13.9 mV and −11.3 mV///CDNVs inhibited the proliferation of NSCLC by suppressing STAT3 activation, generating ROS, promoting cell cycle arrest, and activating the caspase pathway.Exo-An2-siRNA [120]Exosomehuman leukemia monocytic cell line THP-1STAT3 siRNADSPE-PEG_2000_-Mal was added into THP-1 cells after induction into M1 macrophages, and the exosomes were extracted using an exoEasy Maxi Kit. siRNA and An2 were added and incubated.150 nm; −45 mV8.9%/Modified with An2The exosomes could penetrate BBB and target GBM, resulting in superior anti-cancer therapeutic effects.DSPC: 1,2-distearoyl-sn-glycero-3-phosphocholine; DSEPC: 1,2-distearoyl-sn-glycero-3-ethylphosphocholine; DSPG: 1,2-distearoyl-sn-glycero-3-phosphoglycerol; PEG-40: Polyethylene glycol-40; PATO5: Precirol ATO5; C888: Compritol C888; CTAB: Cetyl trimethylammonium bromide; DDAB: Didodecyldimethylammonium bromide; EQ1: Esterquat 1; HSPC: hydrogenated soy L-α phosphatidylcholine; TEOS: Tetraethylorthosilicate; DOAP: 1,2-dioleoyl-3-amino-propane; DOHAP: 1,2-dioleoyl-3-hyaluronan-propane; OCMCS: O-carboxymethyl chitosan; N-2-HACC: N-2-hydroxypropyltimehyl ammonium chloride chitosan; DGPE: 1,2-Dipalmitoyl-sn-glycero-3-phosphoethanolamine; PLA: Poly (L-lactic acid); PEI: polyethyleneimine; HOBt: 1-hydroxybenzotriazole; EDC: 1-[3-(dimethylamino) propyl]-3-ethylcarbodiimide hydrochloride; HMBA: N,N’-hexamethylenebisacrylamide; HFBA: Heptafluorobutyric anhydride.


## 5. Conclusions and Perspectives

Many studies have proved the importance of the STAT3 signal pathway in the initiation, progression, and metastasis of lung cancer. Several drugs targeting STAT3 have achieved remarkable therapeutic effects and are expected to become potential treatments for lung cancer. Different therapeutic strategies, including natural compounds, new small molecule inhibitors, and gene therapies, have shown effective targeting of STAT3. However, it should be pointed out that the low bioavailability of drug candidates and rapid degradation of RNA drugs have limited their application in cancer treatment. In addition, as STAT3 is widely expressed in various cells and tissues to regulate the self-renewal of normal stem cells and to participate in mammary gland development and embryogenesis, the potential adverse effects of STAT3 inhibitors on normal cells cannot be ignored.

Various nanotechnologies have been explored to improve drug solubility, prolong circulation time in vivo, and preferentially deliver drugs to tumor sites through the EPR effect. To date, different types of nanocarriers have been attempted for the targeted delivery of STAT3 inhibitors in the treatment of cancer. Although their results are promising in some preclinical trials, there are still some challenges to be addressed during the process of clinical translation. Firstly, as the application of nanoparticles in the lungs is challenging due to several biological barriers, as discussed above, there is an urgent need to design new and more efficient nanoparticles to achieve ideal therapeutic outcomes. The material, composition, size, and charge of nanoparticles, as well as the cross-talk between tumor and stromal cells, should be carefully considered in order to avoid rapid clearance of nanoparticles, enhance drug penetration, and increase tumor response to therapeutic agents. Secondly, more extensive studies and investigations are necessary to examine the safety and biocompatibility of nanoparticles with long-term metabolism in vivo. Thirdly, as STAT3 activation is involved in drug resistance and immunosuppression, NPs-based co-delivery of STAT3 inhibitor and chemotherapeutic drug/EGFR-TKIs/immune adjuvant is considered a promising strategy for combination therapy. To ensure the safety and synergistic effect of these NPs, it is necessary to carry out detailed and complete studies on their pharmacokinetic, biodistribution, and pharmacodynamic parameters. Fourthly, STAT3 inhibitor-loaded nanosystems have also been prepared for the eradication of LCSCs by targeting the markers (such as CD133 [131] and CD44 [132]) of LCSCs. However, there are no specific markers to accurately identify LCSCs, and it is unclear whether these markers will change in different environments. Moreover, as LCSCs and normal stem cells usually share the same self-renewal signaling pathways, such treatment may also lead to the inhibition of signaling cascades in normal cells and the exhaustion of the normal resident stem cell population. Finally, the preclinical application of most NP-delivered STAT3 inhibitors is currently limited to in vitro cancer cell lines or CDX models. Therefore, it is necessary to validate their enhanced anti-tumor efficacy in more clinically relevant tumor models, such as PDX models, and ultimately translate these preclinical outcomes into clinical applications.

In summary, lung cancer remains one of the most common cancers in both men and women, and the efficiency of conventional treatment has quickly leveled off, so there is an urgent need for more effective treatment strategies. As we have summarized, the STAT3 pathway plays a key role in tumorigenesis, metastasis, drug resistance, and cancer stem cell maintenance of lung cancer. Scientists have developed various intervention strategies with STAT3 as the therapeutic target, including natural compounds, small molecules, and RNA-based therapeutics. They have also attempted to apply multiple nanotechnologies for targeted delivery of STAT3 inhibitors, which significantly improves their bioavailability and tumor targeting capabilities and shows great potential for lung cancer treatment in preclinical trials. We expect that nanotechnology-based targeted delivery of STAT3 inhibitors can simultaneously eliminate both lung cancer cells and cancer stem cells, which will lead to new treatment strategies for lung cancer and will significantly improve patient survival if translated into clinical applications in the near future.

## Figures and Tables

**Figure 1 pharmaceutics-14-02787-f001:**
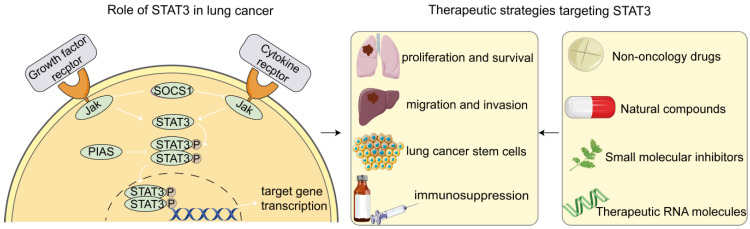
Role of STAT3 in lung cancer and current therapeutic strategies targeting STAT3.

**Figure 2 pharmaceutics-14-02787-f002:**
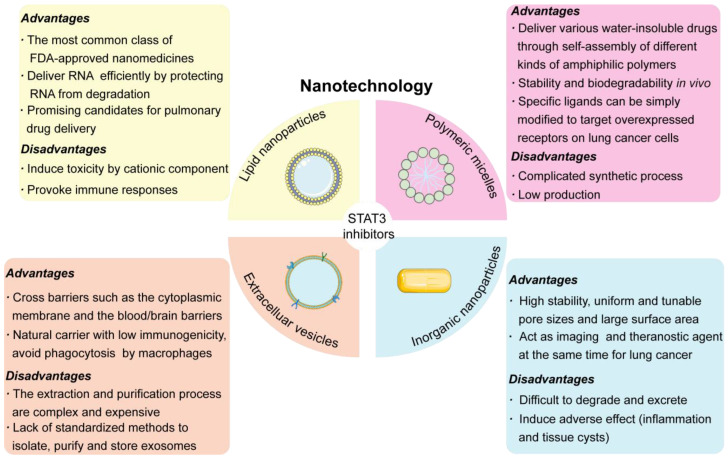
Application of nanoparticles for targeted delivery of STAT3 inhibitors in lung cancer treatment.

**Figure 3 pharmaceutics-14-02787-f003:**
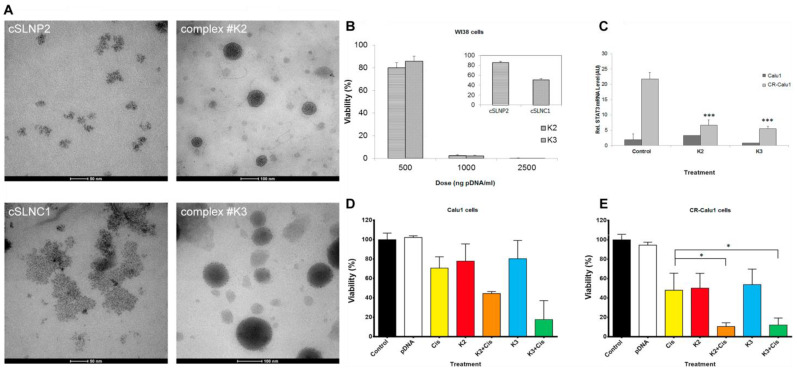
Morphology and cytotoxicity of cSLN:pDNA complexes. (**A**) TEM micrograph of the selected optimal cSLN:pDNA complexes. (**B**) Viability of WI-38 normal lung fibroblast cells treated with different concentrations of complexes (inset viability of WI-38 cells after treatment with the cSLNs of the complexes at an equivalent dose to 500 ng pDNA/mL). (**C**) Relative STAT3 mRNA expression levels in Calu1 and CR-Calu1 cells before and after treatment with cSLN:pDNA complexes. (**D**) Viability of Calu1 cells after treatment with cisplatin, K2, K3, or a combination of complexes with cisplatin at IC_50_ dose. (**E**) Viability of CR-Calu1 cells after treatment with cisplatin, K2, K3, and a combination of complexes with cisplatin at IC_50_ dose. *, *p* < 0.05; ***, *p* < 0.001. (K2: cSLNP2: pDNA, cK3: cSLNC1: pDNA). Reproduced from ref. [81]. Copyright 2017 Elsevier.

**Figure 4 pharmaceutics-14-02787-f004:**
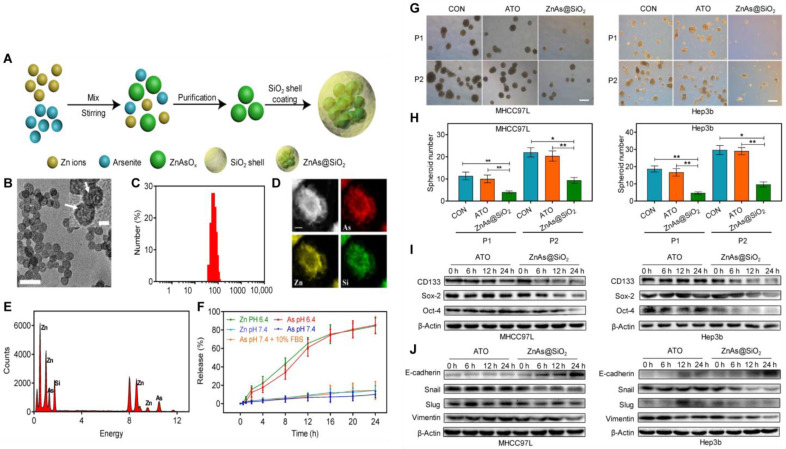
The characteristics of ZnAs@SiO_2_ NPs and their ability to inhibit stemness and EMT in hepatocellular carcinoma cells in vitro. (**A**) Depiction of ZnAs@SiO_2_ NPs synthesis. (**B**) TEM image of ZnAs@SiO_2_ NPs. Scale bar, 50 nm. Insert: the high-magnification TEM image of ZnAs@SiO_2_ NPs indicating the formation of zinc arsenite nano-complexes (white arrows) in the silica shell. Scale bar, 10 nm. (**C**) DLS profile of ZnAs@SiO_2_ NPs. (**D**) Energy-dispersive X-ray mapping images of the ZnAs@SiO_2_ NPs. Scale bar, 1 µm. (**E**) Energy dispersive X-ray spectroscopy of accumulative ZnAs@SiO_2_ NPs in the copper mesh. (**F**) Zn and As ions release at different pH values of 7.4 (with or without 10% FBS) and 6.4 (*n* = 3). (**G**) Typical images of tumor spheroids forming ability of MHCC97L and Hep3b cells. Scale bar, 50 µm. (**H**) Quantification of tumor spheroids forming ability of MHCC97L and Hep3b cells. (**I**) Protein levels of stemness markers in MHCC97L and Hep3b cells. (**J**) Protein levels of EMT markers in MHCC97L and Hep3b cells. *, *p* < 0.05; **, *p* < 0.01. Reproduced from ref. [95]. Copyright 2019 Ivyspring.

**Figure 5 pharmaceutics-14-02787-f005:**
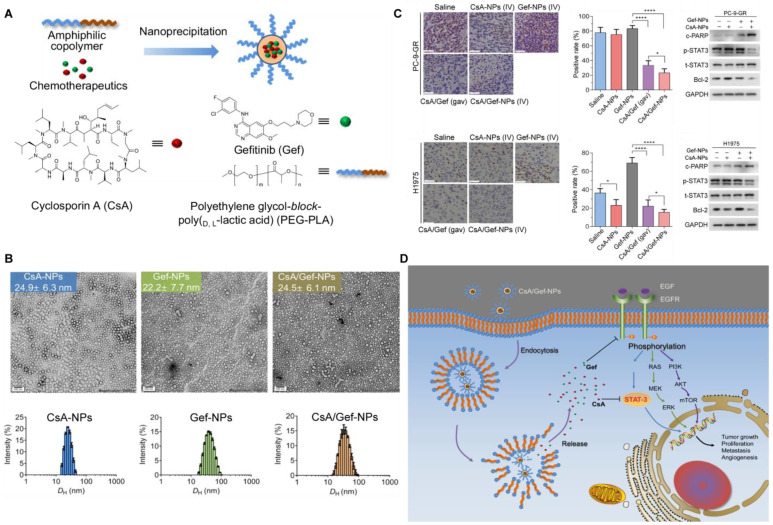
Gefitinib (Gef) and Cyclosporin A (CsA) co-delivered PEG-PLA micellar NPs sensitized Gef-resistant NSCLC to Gef treatment. (**A**) Schematic illustration of the generation of a water-soluble and systemically injectable nanomedicine co-encapsulating two anticancer agents, CsA and Gef, using the amphiphilic block copolymer polyethylene glycol-block-poly(lactide) (PEG-PLA). (**B**) Transmission electron microscopy (TEM) morphology and size distribution of drug-formulated NPs. Scale bars represent 100 nm. CsA inhibits Gef-induced STAT3 activation when co-delivered by NPs in two NSCLC xenograft models. Scale bars represent 50 μm. (**C**) Immunohistochemical (IHC) staining image (left) and quantitative IHC for p-STAT3 results (middle) and Western blot results for p-STAT3/Bcl-2 (right) in PC-9-GR and H1975 xenograft-bearing mice. The magnification of the IHC images is 400×. *, *p* < 0.05; ****, *p* < 0.0001. (**D**) General mechanism explaining how CsA/Gef-NPs overcome multidrug resistance (MDR) and suppress cancer. Reproduced from ref. [105]. Copyright 2018 Springer Nature.

**Figure 6 pharmaceutics-14-02787-f006:**
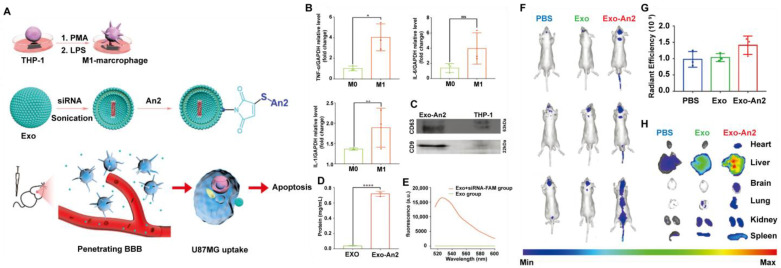
Angiopep-2 (An2) conjugated- (STAT3) siRNA-loaded exosomes (Exo-An2-siRNA) derived from human M1 macrophages could boost BBB permeation and GBM targeting. (**A**) Formation of Exo-An2-siRNA, penetration of the BBB, active targeting, and RNAi therapy in U87MG. (**B**) IL-6, IL-1, and TNF-α mRNA gene expression changed when THP-1 cells were induced into the M1 macrophages. (**C**) Western blotting analysis of CD63 and CD9 in THP-1 cells and Exo-An2. (**D**) The comparison of the amount of An2 in Exo-Mal, Exo-modified An2, and Exo-Mal-modified An2 by BCA. (**E**) Fluorescence spectra of Exo loaded with and without FAM-tagged siRNA. The values represent means ± SD (*n* = 3). * *p* < 0.05, ** *p* < 0.01, **** *p* < 0.0001, ns no significance. (**F**) Fluorescence images of mice bearing U87MG cells treated with PBS, Exo-An2, and Exo (*n* = 3). (**G**) Quantitative statistics of radiant efficiency in the brain of mice. (**H**) Ex vivo fluorescence images of the major organs and major organs excised from mice 24 h after injection. Reproduced from ref. [120]. Copyright 2022 Royal society of chemistry.

## Data Availability

The data in Table 2 are available at http://www.clinicaltrials.gov (accessed on 20 September 2022).

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
