# Peer review of "Nanoparticle-Mediated Delivery of STAT3 Inhibitors in the Treatment of Lung Cancer"

_pharmaceutics, 2022, doi:10.3390/pharmaceutics14122787_

Round 1

Reviewer 1 Report

In this review article, the authors presented "Nanoparticle-mediated delivery of STAT3 inhibitors in the treatment of lung cancer". First, they discussed the role of STAT3 in lung cancer, followed by the possible therapeutic strategies and the different nanoparticle delivery systems for the delivery of STAT3 inhibitors. They have summarized everything via tables and schematic figures. All of these made the article easy to understand. From my point of view, the topic is fascinating. The paper is concise, well-written, and very informative. I recommend the manuscript to publish in Pharmaceutics as is. 

Reviewer 2 Report

In this review, Feng et al. summarized the recent advances in STAT3 targeting strategies, as well as the applications of nanoparticles-mediated targeted delivery of STAT3 inhibitors in the treatment of lung cancer. This review is well-organized and substantial, however, additional clarification and discussion are required before publication. Here are some issues:

1. The compositions of figures in this manuscript are too simple and unrepresentative. Please supplement more related contents in the figures. Further, additional figures are suggested.

2. It is better to review the “Therapeutic strategies targeting STAT3” in more conclusive and inductive statements rather than simply make a list of related works. The small interfering RNAs (siRNAs) is a common STAT3 inhibitor as shown in “Nanoparticle-based delivery of STAT3 inhibitors in the treatment of cancer”, while there is no description about siRNA in “Therapeutic strategies targeting STAT3” section.

3. For the “Nanoparticle-based delivery of STAT3 inhibitors in the treatment of cancer” section, the challenges for lung cancer-targeted delivery of STAT3 inhibitors should be introduced and discussed.

4. The description that " It should be mentioned that since the current application of NPs-based delivery of STAT3 inhibitors in lung cancer is relatively limited, we have also summarized their potential applications in other cancers." is unable to accept. The authors can introduce the barriers to nanocarrier-based lung delivery, the advantages and disadvantages of different nanocarriers in lung-targeted delivery in detail.

Reviewer 3 Report

In the submitted manuscript authors have provided a comprehensive review of nanoparticle-mediated pulmonary delivery of STAT3 inhibitors for cancer treatment.  The article is well organized and presented information is relevant and supported by numerous references. However, what is substantially missing in the manuscript is a critical overview of the presented drug delivery systems in terms of formulation properties (excipients, manufacturing methods, etc.) and the quality attributes that are relevant for delivery of drugs (and bioactive molecules in general) to the lungs. More specifically, there are several hurdles presented by the lungs' anatomy and (pato)physiollogy that need to be carefully addressed. E.g., depending on the desired site of action different aerodynamic properties of nano-sized carriers are desired. And also, different dosage forms may be used for lung administration. Delivery systems' aspects, from the perspective of site of delivery and mechanism of action, need to be taken into consideration at the same time. I agree that some findings for cancer in other tissues and/or organs can be useful as a guidance but please elaborate much more comprehensively the delivery of active compounds to this very specific site of action.

Please correct Figure 2 with “extracellular vesicles” instead of “extracellular vehicles”.

Round 2

Reviewer 2 Report

The authors have significantly revised the paper. All my comments have been addressed. The quality of the review paper has been improved. Thus, I recommend the acceptance of this paper for publishing on Pharmaceutics.  

One small point: The words in the figures are too small. The authors should revise the figures accordingly.

Author Response

Thanks for your suggestions. We have readjusted word sizes in the figures.

Reviewer 3 Report

I think that authors have addressed all comments/issues, therefore the manuscript can be accepted for publication in its current form.

Author Response

Thank you very much.